# Loneliness and Emancipation: A Multilevel Analysis of the Connection between Gender Inequality, Loneliness, and Social Isolation in the ISSP 2017

**DOI:** 10.3390/ijerph19127428

**Published:** 2022-06-17

**Authors:** Janosch Schobin

**Affiliations:** Macrosociology, Department of Social Sciences, Universität Kassel, 34109 Kassel, Germany; jschobin@uni-kassel.de

**Keywords:** gender inequality, loneliness, couples, multilevel analysis

## Abstract

The present article analyzes the connection between, on the one hand, gender equality and, on the other hand, loneliness and social isolation. It hypothesizes that modern relational institutions that support gender equality, such as no-fault divorce laws, reduce loneliness in close relationships. This hypothesis is put to the test through a multilevel analysis of the International Social Survey Program (ISSP) 2017. The analysis reveals that the data agree, to a large extent, with the theoretical arguments. The prevalence of loneliness is higher in countries with higher levels of gender inequality (as measured by the Gender Inequality Index (GII)). This can be attributed to a moderation effect; at lower levels of gender inequality, partnerships provide better protection from loneliness. These results are robust to controls for demographic composition, level of health, educational attainment, income poverty, and interview mode. Last, the analyses show that the threat of emotional isolation is more widespread in countries with low gender inequality. These findings, however, are only significant before controlling for demographic composition, level of health, educational attainment, income poverty, and interview mode, and they require further analysis. The concluding section relates these findings to the popular tendency to argue that modern society has created a “loneliness epidemic” and discusses policy implications.

## 1. Introduction

In recent years, loneliness has become a political issue. So much is evidenced, for example, by the establishment of a Ministry for Loneliness in the UK in 2018 [1,2]. The British example has triggered the institutionalization of anti-isolation policies in other OECD countries. Australian, Japanese, Swedish, and German politicians have shown considerable interest in the British model [3,4]. The political discourse is driven by the perception that the population of contemporary Western societies is becoming increasingly isolated. There is talk of an epidemic, a new widespread disease [5]. This hypothesis is based on known facts; individuals in modern societies increasingly have fewer children, marry less often, divorce more frequently, are more likely to live alone, and they are older on average [6]. Therefore, they are increasingly isolated, and their isolation means that they are lonelier. The present article contends that this argument is flawed by pointing out the positive effect that the rise of gender equality has had on the prevalence of loneliness in modern society. It focuses on female empowerment processes and argues that their influence outweighs the impact that demographic change has had on personal networks and the increase in the proportion of elderly individuals in the general population. By strengthening the autonomy of subjects to shape their relationships, the emancipation processes in modern societies have counteracted one of the primary risk factors for loneliness—poor close relationships. Gender equality reduces emotional isolation in relationships by improving opportunities to choose and conclude relationships. This process has been closely linked to female equality through the establishment of enforceable rights, such as unilateral divorce laws, and the decline of arranged marriage. At the same time, however, the social perception of loneliness has also changed. This might, in part, be attributed to the change of risk perceptions. While progresses in gender equality can reduce the risk of emotional isolation in relationships, they might also sharpen the perception of the threat of emotional isolation due to more concentrated emotional networks. This could explain a contradictory dynamic whereby modern societies are not becoming more lonely but loneliness is being perceived as more threatening by a larger proportion of the population.

The remainder of this article is divided into four sections. The first develops a theoretical argument that explores the mechanism by which modern relational institutions that support gender equality reduce loneliness in close relationships. The Section 2 presents the methodological setup of an empirical study that is based on data from the ISSP 2017. The study inquires whether the theoretical arguments that are presented in the first section agree with evidence from a varied set of cross-national representative surveys on loneliness and social isolation. The Section 3 discusses the results. They show that the data largely cohere with the theoretical arguments. One finding is that the effect of partnerships on loneliness is moderated by the level of gender inequality in a country (as measured by the Gender Inequality Index (GII)). The higher the level of gender equality, the stronger the protective effect of partnerships against loneliness. Furthermore, the study shows that the threat of emotional isolation becomes more widespread as gender equality increases, at least before controlling for demographic composition. Section 5 highlights some limitations of the study and Section 6 relates the findings to the popular tendency to argue that modern society has created a “loneliness epidemic” and discusses policy implications and limitations.

## 2. Theoretical Framework

### 2.1. Gender Equality and Loneliness: Is There an Emancipation Dividend?

Sociological theorists have often made metaphorical use of the word “loneliness” to criticize certain aspects of capitalistic consumer societies, for instance because of the predominant social character that they (supposedly) produce or because of the consumer mentality that they (supposedly) infuse into social relationships [7,8]. Zygmunt Bauman, in a similar fashion, has related consumer society to superficial and individualistic relationship styles that lead to a feeling of empty connectedness, which he terms “loneliness” [9]. Here, a more concrete meaning of the term is adopted, in line with current debates in health and the human sciences, such as gerontology and social psychology, namely, that loneliness is a fundamental change signal, such as hunger or pain, that indicates that an individual’s social connections are too weak or conflict ridden [10]. The studies that adopt this concrete definition of loneliness provide no conclusive evidence for the hypothesis that societal individualization processes lead to an increase in the prevalence of loneliness. This trend in the literature can be exemplified by the ongoing debate on the effect of individualistic cultural orientations on the prevalence of loneliness. The evidence on this idea is mixed [11]. For instance, Swader (2019), in a comparison of 21 European countries, finds that individuals in societies that are more individualistic are less lonely [12]. Other cross-country comparative studies from Western contexts come to similar conclusions [13,14,15], while some find contrary evidence in samples from a wider selection of countries [16,17].

One reason for this inconclusiveness is that the individualism–collectivism hypothesis is based on a set of contradictory pull and push factors that are, for the most part, related to the structure of family relationships [15]. For instance, it has been argued in different ways that collectivistic cultures provide more stable and reliable support from family members, a protective factor. At the same time, higher levels of family cohesion have been related to poorer perceptions of relationship quality, which are due to strong familial obligations and burdens. Thus, the same characteristics of collectivistic societies are expected to protect and to promote loneliness at the same time. Heu et al. (2021) dub these contradictory hypotheses and findings of the individualism–collectivism literature the “cultural loneliness paradox” [18]. They propose to remedy it by inquiring how cultural norms impact opportunities to form new relationships (i.e., relationship mobility), how they regulate the resolution of relationships (i.e., relationship stability), and how the two sets of norms interact. In particular, they find support for the hypothesis that stronger relationship mobility norms are consistently associated with lower levels of loneliness across a sample from four European countries. Weaker relationship stability norms, conversely, were not clearly related to higher levels of loneliness. 

The argument that follows is strongly related to Heu et al.’s (2021) line of thought and applies it to one of the important modernization processes in contemporary societies, namely, the advance of gender equality [18]. As far as loneliness is concerned, gender norms appear to offer a more suitable approach to understanding cultural variation in its prevalence than collectivistic or individualistic orientations. For instance, the increases in women’s labor market participation and female educational attainment are both linked to the postponement of childbearing [19,20], which, in turn, has a decisive causal impact on family structure and thus on the availability of family and kinship relationships. Furthermore, the rise in the equality of women is related to better outcomes in health and mental health [21,22], lower levels of intimate partner violence [23,24], and lower levels of poverty [25,26], all of which are causally connected to experiences of loneliness and/or social isolation [27,28,29,30,31]. 

Furthermore, gender equality is clearly related to the availability of opportunities for relationship mobility. The emancipation process can be reconstructed, in part, as a process of increasing individual control over close relationships. It establishes unilateral systems of relationship dissolution. This process is epitomized by the transformation of divorce law across modern societies from the 1960s onwards [32]; no-fault divorce laws that allow for the unilateral cancelation of marriages have become the norm across modern societies, leading to more divorces. This process has contributed to the shrinking of the group of lonely individuals because, when observed across an entire population of relationships, divorces should have a positive effect on the quality of partnerships. So much is suggested by studies on the subjective well-being and psychological stress of divorcees; before separation, psychological stress is extremely high, and relationship quality is low. After separation, the relationship is lost, but psychological distress decreases sharply and becomes similar to that observed among single individuals [33,34,35]. At the same time, psychological distress among individuals in relationships that endure is particularly low when compared to that observed among non-partnered individuals. This observation can be attributed, in part, to the protective effects of high relationship quality and low relationship stress against loneliness. The selection processes of choosing and leaving partners ensure that, on average, relationship quality increases. In consequence, population-level loneliness decreases. 

Moreover, there is some evidence that supports the additional hypothesis that more gender-equal marriages (and relationships) are more efficient in the production of relational goods such as relationship quality and marital satisfaction [36,37]. This literature suggests that increased gender equality leads to less loneliness not only by producing more supportive, emotionally close, and hence satisfying relationships between men and women through relationship selection processes, but also by supporting more efficient relationship bargaining processes throughout the whole life course of marriages and partnerships. 

The spread of unilateral divorce laws that began in the 1960s may therefore be expected to have reduced loneliness because it made unhappy marriages less likely, not necessarily in the life course of the individual but on the aggregate level of all marital relationships. A similar argument can be made with respect to the diverse and fluid new forms of partnerships and lifestyles that have been spreading in Western industrialized societies since the 1960s. In particular, the progressive acceptance of LGBTQ lifestyles may have prevented feelings of loneliness and social isolation. Several studies have shown that both internalized homophobia and experiences of discrimination on the basis of sexual orientation increase the likelihood of feelings of loneliness and reduce the protective effect of social networks against it [38,39,40]. 

Two objections can be raised against the argument that the emancipation-driven tendency toward greater unilateral control over the dissolution of partnerships has led to lower levels of loneliness. The first is that the increase in the number of lonely singles that divorces and more fluid partnership forms have precipitated outweighs the benefits of preventing loneliness in partnerships. However, this objection is not consistent with most of the evidence from demographic studies. The absolute increase in the number of single households and individuals in contemporary societies reveals little about the frequency of stable partnerships across the lifespan of the average individual. While marriage rates have been declining and divorce rates have been high in the Western world since the late 1960s, these tendencies cannot be attributed conclusively to the notion that exclusive partnerships between two individuals have lost their appeal or have become unattainable for an increasing proportion of the public. To a large extent, the loss of marriages (through divorces and foregone marriages) has been compensated by the spread of legitimate forms of non-marital partnership. In most countries, the rise of cohabitation alone has offset the loss of marriages [41,42]. It has even been disputed that in the course of the decline of marriage, the prevalence of partnerships has decreased substantially beyond what demographic factors can explain [43,44]. For example, it must be considered that women are often partnerless in old age because their partners die before them. If a population contains more aged individuals, the prevalence of singledom increases without a decrease in the lifetime probability of biographical phases of long-term partnerships. This proposition suggests that the increase in loneliness due to greater relationship instability and the consequent failure of the relationship market is marginal in comparison to the increase in the quality of marriages and partnerships around the world that has been precipitated by no-fault divorces and new, more inclusive, and legitimate forms of coupledom. 

The second objection is more difficult to dismiss. It concerns the collateral damage of separation. How do parent–child relationships and the psychosocial resources of children change as a result of divorce or separation? A comprehensive meta-study indicates that divorce has long-term negative effects on the mental health of children [45]. This finding suggests that parental separation is conducive to the development of feelings of loneliness at the population level. While this hypothesis has some merit, it must also be noted that the majority of the studies that the meta-analyses cover does not employ methods that are suitable for isolating the causal effect of divorce on children’s mental health. They usually compare the children of divorced parents with the children of non-divorced parents. Since it can be assumed that parents who divorce differ from parents who do not divorce in respect of the background variables that impact children’s mental health (such as personality traits, mental health, conflict behavior, and attachment style), whether divorce has a negative impact on the mental health of children on average remains open to question. In consequence, the argument for an increase in loneliness is weaker than the statistical associations suggest. Furthermore, Auersperg et al. (2019) show that the adverse mental health effects in question tended to decrease in studies between 1990 and 2017 [45]. This observation suggests that to some extent, the negative effects of divorces can be attributed to a fading divorce culture that impairs children’s relationships to divorced parents, especially fathers [46,47]. This inference, however, would support rather than disprove the hypothesis that advances in gender equality lead to a decline in the prevalence of loneliness. All in all, the literature points to the hypothesis that gender equality and female empowerment should imply a reduction in population-level loneliness because they strengthen the protective effect of average partnerships against loneliness. 

### 2.2. Gender Equality and Social Isolation: The Dynamics of High-Quality, Low-Size Personal Networks

Most researchers of loneliness agree that it has to be differentiated from social isolation [30,48,49], which is understood conceptually as a theoretical analogue to prolonged, intense, and non-self-determined aloneness; an individual is socially isolated if the frequency of contact in their close relationships falls below a certain threshold permanently. It should be noted, however, that there is no scientific consensus on the means of determining this threshold. Various measures have been proposed and see use in current research [27,50,51,52,53]. Furthermore, similarly to the concept of poverty, social isolation is often differentiated into types of “contact poverty”. The classic distinction is that made by Weiss (1973), who distinguishes between emotional and social isolation (to evade the confusion that the homonymy might produce, we call Weiss’ (1973) social isolation “community isolation” hereafter) [54]. Accordingly, individuals are emotionally isolated if the frequency of their contact with primary-group members (such as partners, relatives, and close friends) is severely limited. Individuals suffer from community isolation when their opportunities for socializing in secondary groups (such as clubs, voluntary associations, leisure activity groups, and church congregations) are severely limited. 

The question, thus, is how gender equality is related to social isolation in theory. This question can be framed theoretically; that is, one may inquire how the advance of women’s equality has impacted the availability and structure of personal emotional support networks and social participation in the wider community. One initial observation is that female educational attainment and labor market participation have contributed significantly to the postponement of childbirth and the fall in overall fertility [55]. This effect of female empowerment has causal implications for the typical network structure. In a low-fertility setting, individuals have fewer siblings and, as a consequence, fewer kinship ties [56,57]. The effect on kinship network structures depends strongly on specificities, such as whether a one- or two-or-none child norm becomes dominant [56,57]. This said, a common overall structural effect of low fertility is that the ties that individuals choose, especially ties to partners and, to a smaller extent, to close friends, tend to increase in relative importance in the individual’s core network [58,59]. Moreover, chosen ties in contexts of high gender equality are usually dependent on reciprocal acceptance and are often terminated if the quality of the relationship drops below a threshold that is not acceptable for either party to the relationship. This argument suggests that relationship quality comes at the price of an increase in relationship selectivity, which should reduce the overall number of relationships formed. Additionally, it is also plausible to assume that high quality relationships come at higher “maintenance costs”, i.e., they require more resources such as attention and time to exist, further limiting their number. This line of reasoning suggests that progress in achieving female equality, which is a process of shrinking kin networks and increasing individual control over close relationships, manifests as a concentration of close ties in a smaller number of high-quality relationships. The soft optimization of the core emotional network promotes smaller higher-quality networks. 

What could be argued against this view is that the increase in female labor participation compensates for the shrinking of kinship-based emotional core networks. For instance, Hochschild (1997) found that 47% of women in a survey among 1446 parents of children that attended the Bright Horizons Children’s Centers stated that they had most friends at work [60]. Findings like these suggest that novel opportunities to form close friendships at work more than compensate for the loss of opportunities to form friendships among kin and family, which is the typical locus of friendship formation in more traditional and most historic societies [59]. However, the evidence for this effect is very limited and often suffers from conceptual biases, such as the definition of close friendships as non-kinship relationships, which is a very modern and Western conceptualization that tends to overlook female friendships in more kinship-centered societies. 

The question of workplace acquaintances leads to the question of how to assess the impact of the rise of female equality on community isolation, that is, isolation from secondary groups such as religious communities or political, voluntary, neighborhood, and leisure associations. Here, it is also possible that an increase in the workforce participation of women compensates for a decrease in volunteering or community/church engagement that might follow from less traditional female roles. However, the empirical findings on the impact of social modernization processes such as gender equality on community relationships are notoriously varied. For the USA, for instance, Putnam (2000) argues prominently that societal modernization processes have led to the long-running decline in participation in voluntary civil associations [61]. In Germany, conversely, an increase in participation, accompanied by a shift to more issue-driven and time-limited commitments, has been observed from the 1980s onwards [62]. Religious affiliation has been decreasing for decades in Europe, but it has remained stable in the USA. That secularization is a necessary effect of social modernization processes, such as gains in female equality, was a popular idea in the 1960s and 1970s, but it has since been disproven by empirical facts [63]. Moreover, the evidence about the contradictory notions of “community lost” (modernization processes lead to a reduction in community relations) and “community transformed” (modernization processes lead to new types of community relations that compensate for the loss of traditional community ties) is extremely mixed [64]. These observations, as well as others that are omitted here for brevity, indicate that the social sciences have so far failed to establish a clear link between community isolation and the advance of female equality in society. 

## 3. Hypotheses and Methods

### 3.1. Hypotheses

The present section tries to show that the theoretical arguments about the emancipation dividend and the concentration of emotional networks into higher-quality relationships have empirical merits by examining comparative cross-national data on loneliness and emotional isolation. In Section 3, three empirical hypotheses that can be derived from the theoretical discussion are tested. It is posited, first, that countries with a higher level of gender equality exhibit lower levels of loneliness (Hypothesis 1). The second hypothesis is that the loneliness-reducing effect of having a partner is larger in societies that are more gender equal (Hypothesis 2). This is so because stress in close relationship networks should be lower on average in societies that permit greater relationship mobility, which, as argued above, results from the rise of gender equality (Hypotheses 3). However, gender equality also leads to the contraction of emotional networks because it entails lower birthrates and a stronger emphasis on chosen social ties, such as ties to partners (Hypothesis 4), which leaves a higher number of individuals vulnerable to emotional isolation in more gender-equal societies (Hypothesis 5). The final question that is analyzed concerns the impact of gender equality on community isolation. Significant country-level variation is expected, while the direction of the effect of gender equality is only explored. 

### 3.2. Data

The ISSP 2017 (v2.0.0) was a series of international, comparative, and representative population surveys that were conducted in 30 countries in Europe, North America, South America, Africa, Asia, and Oceania (see the Section 3.4 for more details). The surveys attracted a total of 44,492 respondents. 

### 3.3. Measures

Gender Inequality Index: The GII is an index that varies between 0 (“perfect equality between men and women”) to 1 (“perfect inequality between men and women”). It attempts to capture the degree to which women enjoy social, economic, and political equality in a society through a single measure. It takes into account women’s reproductive health (adolescent births and maternal mortality), their level of education, their labor market participation, and their representation in parliaments [65]. The index was logarithmized and then taken at its absolute value for the purpose of the analyses that are presented. This approach has advantages for the statistical treatments because the GII is strongly left skewed. This can cause problems with outliers in regression models. The absolute value of the logarithm is 1 at 0 and ∞ at 0, meaning that higher values on the absolute log(GII) indicate higher levels of gender equality.

Loneliness: The ISSP 2017 used a short three-item scale to measure the frequency of feelings of loneliness [66]. The items were measured on a five-point frequency scale (1 = “never”; 2 = “rarely”; 3 = “sometimes”; 4 = “often”; 5 = “very often”). The procedure suggested by Taniguchi and Kaufman (2021) was applied to derive a scale from the items [17]. First, the levels “never” and “rarely” were recoded to 0, and the levels “sometimes,” “often,” and “very often” were recoded to 1. In the second step, the respondents were assigned a value of 0 if their total score on the recoded items was 0 (=“not lonely”) and 1 (=“lonely”) otherwise. This procedure yielded an acceptable level of reliability for the total scale (Cronbach’s α = 0.77).

Close-network stress: Close-network stress conceptually refers to the perception of strain, conflict, and burdens that arise in close relationships. It was measured by two items. The first item seeks to capture perceptions of strain due to exigencies that originate from family members, relatives, and friends: “Do you feel that your family, relatives and/or friends make too many demands on you?” (1 = “No, never”; 2 = “Yes, but rarely”; 3 = “Yes, sometimes”; 4 = “Yes, often”; 5 = “Yes, very often”). The second item refers to perceptions of emotional stress in a respondent’s close network. The question that was asked was as follows: “Thinking about the important people in your life, such as your spouse or partner, your family members, or close friends, how often in the past 4 weeks did any of these people act angry or upset with you? (1 = “never”; 2 = “rarely”; 3 = “sometimes”; 4 = “often”; 5 = “very often”). The total scale had a Cronbach’s α of 0.57, which is poor but still acceptable given the small number of items. The means of both items were taken to build the close-network stress index ${\mathrm{CNI}}_{\mathrm{stress}}.$

Emotional isolation: The ISSP 2017 used two types of measures to assess the inclusion of a participant into a network of primary relationships. The presence of a partner or spouse (0 = “no partner/spouse”; 1 = “has partner/spouse”) and the number of children in a household were measured in discrete units, while the presences of adult children, parents, siblings, more distant family members, and close friends were measured by the frequency of contact with the respondent (1 = “daily or lives in the same household”; 2 = “several times a week”; 3 = “once a week”; 4 = “two or three times a month”; 5 = “once a month”; 6 = “several times a year”; 7 = “less often”; 8 = “never, is not alive, or does not apply”). These variables were first grouped into four segments: parents, children (children in household (HH) and adult children), kin (siblings and more distant family members), and close friends. To ensure their comparability, the variables were recoded using the following scheme: the variable “number of children in HH” was dichotomized (0 = “no children in HH”; 1 = “children in HH”). The frequency scales were dichotomized at the value “several times a week” (1 = “smaller”; 3, else 0). This approach amounts to considering a set of close relationships to be present in a respondent’s core emotional network if the contact that they provided was at a comparable level to social contact with someone living in the same household. The following social network index was thus calculated: ${\mathrm{SNI}}_{\mathrm{emo}}=2\times \mathrm{Partner}+\mathrm{Parents}+\mathrm{Children}+\mathrm{Kinship}+\mathrm{Close}\;\mathrm{Friendship}.$ The scale ranges from 0 (“complete emotional isolation”) to 6 (“strong emotional inclusion”). An individual is considered at risk of emotional isolation if their ${\mathrm{SNI}}_{\mathrm{emo}}$ is greater than 1 but smaller than 3, that is, when losing contact with one additional element of their close relationship network would produce emotional isolation. Likewise, an individual is considered emotionally isolated if their ${\mathrm{SNI}}_{\mathrm{emo}}$ is smaller than 2 (The SNI_emo_ has to be considered as a formative index, because it is composed of components that by theoretic construction should contribute independently to the construct we want to measure and do not represent a “symptom” thereof. Reliability measures for reflective scales, such as Cronbach’s α, therefore do not apply. Note that partnerships have a weight of two. In their seminal paper, Berkman and Syme (1979) argue strongly that the relationships summed up in a (formative) social network index should be weighted relative to their importance for social support in order to be a valid measure of network integration [55]. While the relevance of different types of close relationships certainly varies at the individual level, in general, it is accepted that partnerships tend to be the most central and relevant for social support for most people, if they have them. For this reason, partners were given a weight equal to the other two personal-network segments assessed (family members and friends/kin) to reflect their typical high relevance for all types of social support). 

Community isolation: The ISSP 2017 used three items to measure the inclusion of a participant into a network of secondary relationships. It measured inclusion in leisure, sportive and cultural activities, and political associations as well as participation in voluntary work, religious organizations, and charities. The following questions were asked: “In the past 12 months, how often, if at all, have you taken part in activities … (a) of groups or associations for leisure, sports or culture; (b) of political parties, political groups, or political associations; and (c) of charitable or religious organizations that do voluntary work?”. The items were measured on a five-point frequency scale (1 = “once a week or more”; 2 = “one to three times a month”; 3 = “several times in the past year”; 4 = “once in the past year”; 5 = “never”). The frequency scales were dichotomized at the value of “once a week or more” (i.e., “once a week or more” = 1, 0 otherwise). This approach amounts to considering a community relationship to be present if it provides a similar level of contact as a close emotional relationship. The following social network index was calculated from the three items: ${\mathrm{SNI}}_{\mathrm{community}}=\mathrm{Leisure}+\mathrm{Politcs}+\mathrm{Charity}.$ The scale ranges from 0 (“community isolation”) to 3 (“strong community inclusion”).

Control variables: The ISSP 2017 included a variety of control variables that have been connected causally to loneliness and social isolation, such as age, gender, subjective health, depressiveness, educational attainment, income poverty, employment status, and interview mode [27,67,68,69,70,71,72,73,74]. Age was measured in full years. The “age” variable was transformed by cubic orthogonal polynomials to accommodate the multimodal shapes of age dependency that are often found in the literature on cross-sectional surveys, loneliness, and social isolation [75,76,77]. Gender was measured by two categories (0 = “male”; 1 = “female”). Subjective health was captured on a five-point scale (1 = “excellent”; 2 = “very good”; 3 = “good”; 4 = “fair”; 5 = “poor”), which was intended to capture the general state of the respondent’s health. Depressiveness was assessed through the question, “During the past 4 weeks how often have you felt unhappy and depressed?”. The ordinal format of the responses includes five frequencies (1 = “never”; 2 = “rarely”; 3 = “sometimes”; 4 = “often”; 5 = “very often”). Subjective health and depressiveness were interpreted as continuous predictors and normalized for the purpose of statistical analysis. The highest educational degree that a respondent had attained was used to measure their level of education. The responses were homogenized to seven ordinal levels to facilitate cross-national comparability (0 = “no formal education”; 1 = “primary school”; 2 = “lower secondary”; 3 = “upper secondary”; 4 = “post-secondary, non-tertiary”; 5 = “lower-level tertiary”; 6 = “upper-level tertiary (Master, Doctor)”). The following question was used to assess income poverty: “Thinking of your household’s total income, including all the sources of income of all the members who contribute to it, how difficult or easy is it currently for your household to make ends meet?”. The original response format had five levels, which were recoded into a binary variable that indicates whether a respondent found it difficult to cope at their current level of household income (1 = “very difficult or fairly difficult”; 0 = “neither easy nor difficult, fairly easy, or very easy”). Finally, employment status was captured by three categories (0 = “employed”; 1 = “unemployed”; 2 = “not working”), and interview mode was captured by five (1 = “paper and pencil face to face interview (PAPI)”; 2 = “computer aided face to face interview (CAPI)”; 3 = “self-administered paper and pencil questionnaire (SC)”; 4 = “computer assisted web interview/computer assisted self-interview (CASI/CAWI)”; 5 = “telephone and other”).

### 3.4. Sample Description 

The sample contained data from 30 countries that vary substantially in their level of gender inequality, the prevalence of loneliness (Loneliness), close-network stress (${\mathrm{CNI}}_{\mathrm{stress}}$), emotional network integration (${\mathrm{SNI}}_{\mathrm{emo}}$), emotional isolation or risk thereof (fraction of individuals with ${\mathrm{SNI}}_{\mathrm{emo}}$ lower or equal to 2 points), and community network integration (${\mathrm{SNI}}_{\mathrm{community}}$). In terms of gender equality, Table 1 shows that the values on the GII ranged from very low gender inequality (0.040) in Denmark (DK) to very high levels of gender inequality (0.524) in India (IN). The average country displayed a medium level of gender inequality (M = 0.172). The prevalence of loneliness varied from 0.065 points (or 6.5%) in Thailand (TH) to 0.275 (or 27.5%) in South Africa (ZA). The country-level average was 0.175 (or 17.5%). The level of close-network stress, as measured by ${\mathrm{CNI}}_{\mathrm{stress}}$, ranged from 1.564 points in Austria (AT) to 2.623 points in Slovakia (SK). The average level of 1.920 on the country level pointed to medium-to-low average levels of close-network stress. The average of the country averages on the ${\mathrm{SNI}}_{\mathrm{emo}}$ was 3.280. The average fraction of individuals who were emotionally isolated or at risk was 30.0%, pointing towards a normality of stable emotional network inclusion for a large majority of the individuals in most of the countries in the sample. In contrast, ${\mathrm{SNI}}_{\mathrm{community}}$ only ranged between 0.064 points in Lithuania (LT) and 0.449 points in India, averaging 0.251 across all countries. Community integration, as measured here, appears to have been rather weak across the countries in the sample. For further information on the sample, an in-depth country-level description of data is given by Hadler et al. 2020 [78], and a table describing the socio-demographic composition of the sample is included in Appendix A (Table A6).

### 3.5. Missing Data Imputation

From the initial 44,492 respondents of the ISSP 2017, 3.00% (3.80%, 4.77%, 5.21%) had at least one missing value for an item from the Loneliness scale (${\;\mathrm{CNI}}_{\mathrm{stress}},{\;\mathrm{SNI}}_{\mathrm{emo}},{\;\mathrm{SNI}}_{\mathrm{community}})$, and 1.23% (1.30%, 0.34%, 1.47%) were missing all values for those items. To reduce the biases that would be associated with dropping a substantial number of missing cases [79], which is associated with selective item non-response, the following procedure was adopted: cases that had missing values for all items of a dependent variable were eliminated. In cases that exhibited selective item non-response, the corresponding items were imputed multiple times. Furthermore, the non-response rate for control-variable items was moderate (2.44% on average across all eight control covariates that were considered). However, in two cases, the non-response rate was associated with individual countries. In the case of Denmark, all information on income poverty was missing, and information on employment status in South Africa was wholly absent. In order to avoid dropping information from whole countries, all missing data on the control covariates were imputed jointly with the selectively missing data on the dependent variables. Accordingly, 20 imputations by chained equation, using random forests, were calculated following the procedure proposed by Stekhoven and Bühlmann (2012) in order to impute the missing data [80]. All imputations were carried out using the R package *miceRanger* v.1.5.0 [81]. This procedure yielded a total of 43,269 complete cases in each of the 20 imputed datasets. 

### 3.6. Analytical Strategy 

Three nested multilevel general linear models were considered and compared for each dependent variable (Loneliness, ${\mathrm{CNI}}_{\mathrm{stress}},{\;\mathrm{SNI}}_{\mathrm{emo}},\;\mathrm{Emotional}\;\mathrm{Isolation},{\;\mathrm{SNI}}_{\mathrm{community}}$), and the type of link function (identity or logit) was chosen depending on the type of the dependent variable (continuous or binary). The first step of the modelling involved a baseline model. It consisted of random intercept models that account for the country-level variation of the dependent variable. The second step assessed whether between-country variation was attributable, in part, to variations in gender equality; that is, it included the absolute value of the logarithm of the GII (|logGII|) as a country-level predictor. The third group of models extended the second model through individual-level controls for age, gender, subjective health, depressiveness, employment status, income poverty, educational attainment, and interview mode to ascertain whether the country-level association of the dependent variable with gender inequality is attributable to individual-level factors. The variables were selected based on two criteria: (1) They that have been either linked causally to the dependent variable (e.g., depressiveness and health), or (2) they are routinely included in analyses to rule out sample composition effects due to differences in the sampling strategy deployed (e.g., age and gender). In the case of Loneliness, a fourth model was estimated. It includes the effect of partnerships on loneliness and the cross-level interaction of partnerships with gender equality in order to evaluate Hypothesis 2, which proposes that the protective effects of partnerships increase in more gender-equal societies. The nested models were estimated by Restricted Maximum Likelihood (REML) and compared to the baseline model through likelihood ratio tests after being refitted using Maximum Likelihood (ML). The estimations based on the 20 imputation datasets were combined in accordance with Rubin’s rules [82]. 

## 4. Findings

### 4.1. Loneliness, Partners, and Close-Network Stress 

The prevalence of loneliness, as measured here, varied substantially between countries. Table 2 shows that the (adjusted) intra-class correlation (ICC) of the base model (only country-level random intercepts) showed that approximately 5.9% of the total variance could be attributed to the country level. Introducing the absolute log of the GII into the base model as a fixed effect reduced this figure significantly to 5.1% (see Model 1). (Models containing the fixed effects parameters of all control variables are included in Appendix A). Moreover, the analyses of the (discrete) marginal effects of the base model at representative values (MERVs) showed that an increase in the value of |logGII| from 1 (low gender equality, corresponding approximately to a GII of 0.37, roughly the level of gender inequality in countries like Thailand or South Africa) to 2 (middle gender equality, corresponding approximately to a GII of 0.14, roughly the level of gender inequality in countries like New Zealand or Lithuania) entailed a decrease in the fraction of lonely respondents of 0.034 points (or 3.4%). A further one-point increase in |logGII|, to a value of 3 (high gender equality, corresponding approximately to a GII of 0.05, roughly the level of gender inequality in countries like Finland or Denmark) predicted a decrease in the fraction of lonely individuals of 0.030 points (or 3%). These general findings were robust to fixed effect controls for age, gender, educational attainment, poverty, employment status, depressiveness, subjective health, and interview mode (see Model 2). In the model that includes controls (Model 2), the variance attributable to the country level decreased substantially to about 3.8% (adjusted) or 2.8% (conditional, i.e., when the variation attributable to the fixed effects was taken into account). The controls also reduced the MERV estimates for changes from low to medium gender equality (0.027 points, or a 2.7% reduction) and from medium to high gender equality (0.023 points, or a 2.3% reduction). While this finding shows that the correlation between gender equality and the prevalence of loneliness is attributable to differences in the country-level prevalence of individual level factors associated with loneliness to a certain extent, it also lends support to the hypothesis that the level of loneliness decreases with an increase in the gender equality of a country (Hypothesis 2). Moreover, Model 3 revealed that the level of gender equality moderated the loneliness-reducing effects of partnerships significantly, which is concordant with the predictions of Hypothesis 3: the higher the level of gender equality, the stronger the protection that partnerships provide against loneliness.

Figure 1 compares three scenarios to illustrate the results from Model 3. In a low gender equality context (|logGII| = 1), having a partner was associated with a decrease in the prevalence of loneliness of 0.034 points (or 3.4%). In contrast, in medium (|logGII| = 2) and high (|logGII| = 3) gender equality contexts, having a partner reduced the probability of feeling lonely by 0.061 (or 6.1%) and 0.080 (or 8.0%), respectively. In conclusion, the higher the level of gender equality, the stronger the protection that partnerships afford against loneliness. This conclusion must be analyzed in the light of close-network stress. Here, the base model in Table 3 (Model 4) attributed about 10.1% of the variation in ${\mathrm{CNI}}_{\mathrm{stress}}$ to the country level. This proportion was reduced substantially by introducing gender equality as a country-level fixed effect, by 7.3% (ICC adjusted) or 7% (ICC conditional), suggesting that the country-level variance of close-network stress is associated significantly with gender equality. The linear-regression Model 4 showed that a one-unit increase in the absolute logarithm of the GII corresponded to a reduction in the level of close-network stress, as measured by the ${\mathrm{CNI}}_{\mathrm{stress}}$ scale, of 0.197. This result was robust to controls for age, gender, educational attainment, poverty, employment status, depressiveness, subjective health, and interview mode (see Model 5, Table 3). All in all, these results support the hypothesis that the level of close-network stress decreases as gender equality in a country increases (Hypothesis 3).

### 4.2. Emotional and Community Isolation 

As far as the diversity of an emotional network and its size are concerned, the base model in Table 4 (Model A) showed that approximately 7.4% of the variation in ${\mathrm{SNI}}_{\mathrm{emo}}$ was attributable to the country level. This proportion was reduced significantly, to 6.6% (ICC adjusted) or 6.5% (ICC conditional), by introducing gender equality as a country-level fixed effect. The multilevel linear regression (Model A) showed that a one-unit increase in the absolute logarithm of the GII corresponded to a significant 0.208-point fall in ${\mathrm{SNI}}_{\mathrm{emo}}$. This result was robust to controls for age, gender, educational attainment, poverty, employment status, depressiveness, subjective health, and interview mode (see Model B, Table 4). The finding is congruent with Hypothesis 3, which states that progress in gender equality leads to smaller emotional networks of higher quality. However, the concentration of emotional networks does not automatically prove that there is an increase in the number of individuals who are emotionally isolated (or at risk) in more gender-equal societies. For instance, it is possible that the concentration of the emotional network is primarily a product of the elimination of highly redundant sources of emotional support, such as kin or family relationships, which would lead to an increase in the proportion in the middle levels of the ${\mathrm{SNI}}_{\mathrm{emo}}$ (values between 3 and 5) but not in the lower levels (values between 0 and 2). In this case, no increase in emotional isolation would result from a fall in the number of ties. 

In order to investigate this possibility further, a multilevel logistical model was estimated to determine whether the proportion of individuals with low values of emotional network inclusion (${\mathrm{SNI}}_{\mathrm{emo}}$ < 3) increases with gender equality. An analysis of the MERVs of a model that only included country-level random effects and country-level fixed effects for gender equality (Model C, Table 5) suggested that this is the case, which would be consistent with Hypothesis 4: an increase from low (middle) to middle (high) gender equality entails an increase of 4.0% (4.4%) in the fraction of individuals who are emotionally isolated or at risk of isolation. However, the evidence that the ISSP 2017 provided was not robust to controls for age, gender, educational attainment, income poverty, employment status, depressiveness, subjective health, and interview mode. The statistical association between country-level gender equality and individual emotional isolation (or risk thereof) ceased to be significant. It should be noted that the predictions of the model with controls (Model D, Table 5) were not qualitatively different from the results of the simpler model. Therefore, while the data provided some support for Hypothesis 4, the finding should not be considered conclusive. The suggestion is that the observed positive correlation between emotional isolation and gender equality is mediated by changes in demographic composition that are caused by ageing (lower fertility, higher age, etc.). These changes, however, can in part be attributed causally to increases in gender equality on the societal level [19,55], so that the controls in this case would be masking the indirect effect of gender equality. Disentangling this issue through causal analysis techniques would require more than cross-sectional data (e.g., panel data or data that contains suitable instrumental variables). 

Last, the exploratory results on the connection between community isolation and gender equality did provide some tentative guidance for further research. The base model in Table 6 (Model I) showed that a significant proportion (5.3%) of the variation in ${\mathrm{SNI}}_{\mathrm{community}}$ could be attributed to the country level. This outcome reflected expectations from the literature review. Introducing gender equality into this model as a country-level fixed effect yielded the observation that community isolation seems to diminish when gender equality increases. A one-unit increase in |logGII| entailed an increase of approximately 0.045 points in ${\mathrm{SNI}}_{\mathrm{community}}$ (see Model I, Table 6). Thus, in general, it seemed that more gender-equal societies exhibited higher levels of community integration in the ISSP 2017. However, this finding was not robust to controls for age, gender, educational attainment, poverty, employment status, depressiveness, subjective health, and interview mode (see Model II, Table 6). That said, the predictive results were essentially equivalent to the results from the constrained model (Model I, Table 6). All in all, the ISSP 2017 appeared to indicate that the hypothesis that there is a negative relationship between community isolation and gender equality is more likely to hold than its negation. However, as with the fraction of emotionally isolated individuals, the data suggested that the connection is mediated by individual-level traits, and further investigation through more complex data and analytical techniques would be warranted. 

## 5. Discussion

In contrast to the literature on collectivistic and individualistic orientations, the present paper focused on the idea that between-country variance in the prevalence of loneliness and social isolation can be explained by differences in the level of gender (in)equality. For the most part, the analyses that were presented confirm the hypotheses that derive from this approach: more gender-equal societies tend to exhibit a lower incidence of loneliness. This tendency is consistent with lower levels of close-network stress and, accordingly, with a stronger protective effect of partnerships against loneliness. Conversely, societies with higher levels of gender equality exhibit more concentrated emotional networks and higher proportions of individuals who are emotionally isolated or at risk of emotional isolation. However, the statistical significance of the latter association is not robust to controls for the individual-level factors associated with social isolation, such as social status variables (educational attainment, income poverty, and employment status), sociodemographics (age and gender), health outcomes (subjective health and depressiveness), and interview mode. Lastly, the analyses of the data from the ISSP 2017 suggest that it would be desirable to explore the hypothesis that gender equality is likely to be related to increases in community embeddedness through more frequent participation in voluntary work, leisure activities, and religious communities. 

However, the present research and its conclusions are subject to several limitations. Most pressingly, it must be noted that the evidence presented here is cross-sectional, but the hypotheses considered are dynamic. For instance, Hypothesis 1 expects a reduction of the level of gender inequality to entail a reduction in the prevalence of loneliness. Repeated cross-national cross-sectional and/or panel data would be better suited to provide support for this idea than cross-sectional data alone. The present research, therefore, only provides a first step toward establishing a negative temporal association between loneliness and gender equality on the societal level. Further, in terms of establishing the causality of the connection between loneliness and gender equality, the general limitations of observational data must be considered. These can, at least in theory, be addressed by adjusting for variables that block all backdoor paths through which gender inequality and loneliness/social isolation are associated statistically. The present research tries to approximate this procedure by controlling for variables that are believed to be causally associated with loneliness/social isolation, such as subjective health, depressiveness, or educational attainment. However, this procedure could still suffer from simultaneity biases, for instance, if the causalities with the dependent variable are reciprocal, or from omitted-variable biases if important variables that co-determine gender inequality and loneliness are not considered. Expanding the current research in the direction of longitudinal data and enriching or combining them with data from lab, field, and natural experiments could address these methodological shortcomings in future research. A further limitation stems from the selected number of countries considered. Sample-selection processes can produce statistical associations between unrelated variables (collider biases). While it is not clear that this is the case for the ISSP data, it cannot be ruled out either. Moreover, the selection of a small number of countries also limits the possibility to control for other country-level outcomes, such as the level of corruption or the overall level of violence in a society, that might be correlated with but conceptually different from gender equality. Last, the issues concerning the cultural diversity of loneliness concepts could not be addressed by the present research. Cultural differences could impact the measurement of loneliness substantially and invalidate the conclusions drawn from the data. For instance, it is possible that the level of stigmatization of expressing loneliness varies with gender equality. This would produce a correlation between gender equality and loneliness on the societal level without implying a difference in the prevalence of actual feelings of loneliness. In addition, the measurement of social isolation in the ISSP 2017 is rather coarse. It might omit important relationships. Therefore, it is possible that it does not capture social isolation equally well across contexts of different levels of gender equality. This could bias or invalidate the conclusions. However, the ISSP 2017 data do not present a clear way to address issues like these. 

## 6. Conclusions

Notwithstanding the multiple limitations of the present research, the complete set of findings is compatible with the hypothesis that gender equality has contributed to a reduction of loneliness in modern societies by producing a social framework that is more conducive to maintaining high-quality close relationships. From a wider perspective, this has important policy implication because it suggests a shift in the risk profile that typically leads to loneliness in modern societies. Psychological studies draw an important distinction between loneliness caused by the absence of relationships and loneliness caused by poor relationship quality [83,84]. The analyses presented here suggest that in more gender-equal societies the proportion of individuals that experience loneliness because of the absence of close relationships grows, while the proportion that endure loneliness due to poor close relationships diminishes. If this conjecture is valid, it has important policy implications. In particular, it suggests that in social contexts of high gender equality, social policies to prevent loneliness should focus on the institutions and dynamics that secure the efficient functioning of marriage, partnership, and friendship “markets”. Moreover, the present research underscores the importance of life-course-sensitive prevention policies that focus on problems with transitions around major life events. Separation, divorce, bereavement, moving, or changes in employment should be considered situations in which individuals in high-gender-equality societies are at a higher risk of starting a trajectory toward loneliness and social isolation [85]. Allocating more funds to policies that focus on supporting individuals during such transitions could, therefore, be a good starting point to prevent loneliness in high-gender-equality contexts. Finally, in high-gender-equality societies, the issue of the special group of socially isolated lonely individuals becomes more pressing. This group can be assumed to be harder to reach due to its social isolation and special in its needs due to the loss of interpersonal trust and trust in institutions [86,87]. Therefore, special policies must be tailored to its needs. Social prescribing and the development of special connector services can be seen as innovations that are starting to fill this gap [88]. 

In contrast, in low-gender-equality contexts, social policies that primarily aim at reducing and preventing loneliness are probably more effective if they focus their resources on factors that reduce social stressors such as poverty and intimate violence that put strains on close relationships. Moreover, policies that support the individual’s material dependence on kinship and family, their educational opportunities, and the equality of opportunities between genders in general should be considered win–win policies; in addition to preventing loneliness, they also foster other societal aims, such as improving public health and economic prosperity. 

The present research opens several avenues for further study. First, as is apparent from the limitations, an extension to longitudinal analyses of cross-country data and to non-observational data seems warranted. Moreover, further studies similar to Hudiyana et al., 2021 that investigate whether cultural differences bias loneliness measurements are important to establish a firm basis for future cross-national comparative research [89]. A critical issue for examination is the heterogeneity of the effect of gender equality on loneliness. The present research examined it from the perspective of whole populations. However, this could conceal obvious heterogeneities on the individual level as well as on the level of social subgroups. The most obvious extension is gender itself; it seems plausible, for example, that the reduction in loneliness would be more pronounced for women than for men because the privileges enjoyed by men in less gender-equal societies favorably influence their perception of the quality of their partnerships. However, the effects of gender equality might also vary across other aspects of social identity, such as socio-economic status, class, race/ethnicity, and religious affiliation. For instance, it could be that the emancipation dividend is more of a “middle-class luxury” that poorer, working-class women cannot afford. Likewise, it is possible that some groups in the population “lose out” when gender equality increases. An obvious group to consider here is young, low-social-status men who are disadvantaged in relationship markets. Social phenomena like the incel movement (Incel stand for involuntary celibate) point toward considering this. Research along these lines should seek to improve the understanding of the impact of social isolation and loneliness on modern societies. 

## Figures and Tables

**Figure 1 ijerph-19-07428-f001:**
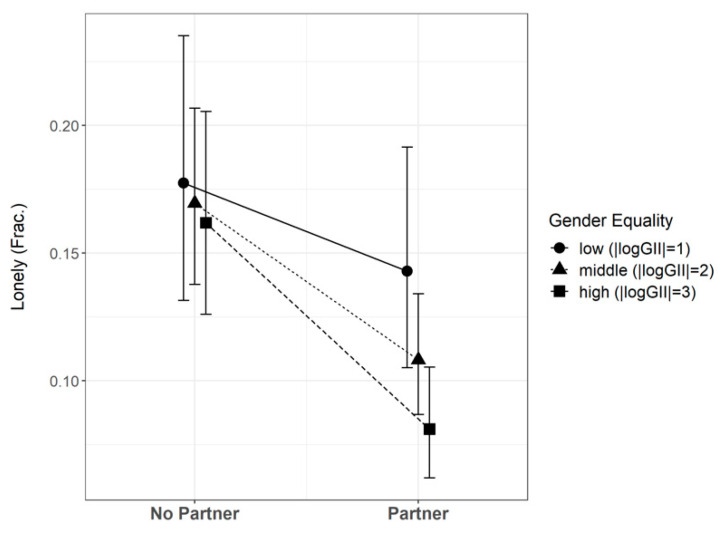
Marginal effects of partnership on loneliness at representative levels of gender equality (based on Model 3).

**Table 1 ijerph-19-07428-t001:** Country-level descriptive statistics.

Country	GII 2017	|log(GII 2017)|	Lonely (Frac.)	CNI__stress_	SNI__emo_	Emotionally Isolated or at Risk (Frac.)	SNI__community_	N
AT	0.071	2.645	0.107	1.564	3.134	0.354	0.275	1199
AU	0.109	2.216	0.263	1.943	2.88	0.378	0.372	1246
CH	0.039	3.244	0.079	1.615	3.341	0.26	0.411	1064
CN	0.152	1.884	0.155	1.596	3.159	0.312	0.072	4199
CZ	0.124	2.087	0.186	2.082	2.988	0.382	0.176	1400
DE	0.072	2.631	0.122	1.62	3.284	0.288	0.349	1661
DK	0.04	3.219	0.151	1.79	3.022	0.331	0.385	1020
ES	0.08	2.526	0.145	1.661	3.837	0.181	0.213	1725
FI	0.058	2.847	0.235	2.15	2.966	0.35	0.243	1047
FR	0.083	2.489	0.206	1.792	3.012	0.345	0.347	1427
GB-GBN	0.116	2.154	0.245	1.864	2.697	0.44	0.334	1523
HR	0.124	2.087	0.208	2.159	2.715	0.363	0.267	998
HU	0.259	1.351	0.214	1.864	3.095	0.376	0.088	1005
IL	0.098	2.323	0.153	1.898	4.002	0.159	0.35	1245
IN	0.524	0.646	0.258	2.142	4.142	0.113	0.449	1462
IS	0.062	2.781	0.19	1.736	3.757	0.196	0.326	1359
JP	0.103	2.273	0.134	1.994	2.506	0.516	0.117	1509
LT	0.123	2.096	0.149	1.787	2.925	0.421	0.064	1015
MX	0.343	1.07	0.142	2.347	3.633	0.212	0.251	986
NZ	0.136	1.995	0.25	1.927	3.186	0.303	0.439	1322
PH	0.427	0.851	0.222	1.858	3.82	0.188	0.122	1182
RU	0.257	1.359	0.13	2.033	3.478	0.274	0.114	1529
SE	0.044	3.124	0.173	1.778	3.283	0.268	0.289	1104
SI	0.054	2.919	0.069	1.732	3.589	0.23	0.277	1046
SK	0.18	1.715	0.197	2.623	3.098	0.344	0.193	1399
SR	0.441	0.819	0.206	2.284	3.605	0.238	0.229	1031
TH	0.393	0.934	0.065	2.504	3.997	0.164	0.261	1425
TW	0.056	2.882	0.11	1.705	2.911	0.395	0.126	1949
US	0.189	1.666	0.222	1.803	3.216	0.314	0.238	1168
ZA	0.389	0.944	0.275	1.875	3.244	0.314	0.145	3024

**Table 2 ijerph-19-07428-t002:** Multilevel logistic regressions of the effect of GII on loneliness.

	Model 3			Model 2			Model 1		
Predictor	Estimate	SE	Sig.	Estimate	SE	Sig.	Estimate	SE	Sig.
Intercept	−1.438	0.276	***	−1.108	0.267	***	−1.115	0.221	***
|logGII|	−0.055	0.109		−0.222	0.104	**	−0.223	0.101	**
|logGII| * Partner	−0.263	0.04	***						
Partner	0.006	0.082		−0.497	0.031	***			
Controls	Yes			Yes			No		
SD Country	0.369	0.005		0.36	0.005		0.421	0.002	
ICC adjusted	0.04			0.038			0.051		
ICC conditional	0.029			0.028			0.051		
LRT-Test	Chisq = 7129.69 DF = 22 Pr(>Chisq) < 0.001			Chisq = 7084.16 DF = 21 Pr(>Chisq) < 0.001			Chisq = 4.34 DF = 1 Pr(>Chisq) < 0.0372		

Signif. codes: “***” *p* < 0.001, “**” *p* < 0.01, “*” *p* < 0.05.

**Table 3 ijerph-19-07428-t003:** Multilevel linear models of effect of GII on SNI_stress_.

	Model 5				Model 4	
Predictor	Estimate	SE	Sig.	Estimate	SE	Sig.
Intercept	2.343	0.136	***	2.336	0.117	***
|logGII|	−0.215	0.057	***	−0.197	0.053	***
Controls	Yes			No		
SD Country	0.225	0.001		0.222	<0.001	
SD Observation	0.745	<0.001		0.794	<0.001	
ICC adjusted	0.084			0.073		
ICC conditional	0.071			0.07		
LRT-Test	Chisq = 5563.27 DF = 20 Pr(>Chisq) < 0.001			Chisq = 11.91 DF = 1 Pr(>Chisq) < 0.001		

Signif. codes: “***” *p* < 0.001.

**Table 4 ijerph-19-07428-t004:** Multilevel linear models of effect of GII on SNI_emo_.

	Model B				Model A	
Predictor	Estimate	SE	Sig.	Estimate	SE	Sig.
Intercept	3.342	0.235	***	3.71	0.208	***
|logGII|	−0.191	0.098	**	−0.21	0.095	**
Controls	Yes			No		
SD Country	0.38	0.004		0.394	0.001	
SD Observation	1.404	<0.001		1.485	<0.001	
ICC adjusted	0.068			0.066		
ICC conditional	0.061			0.065		
LRT-Test	Chisq = 4874.42 DF = 20 Pr(>Chisq) < 0.001			Chisq = 4.81 DF = 1 Pr(>Chisq) < 0.0283		

Signif. codes: “***” *p* < 0.001, “**” *p* < 0.01.

**Table 5 ijerph-19-07428-t005:** Multilevel logistic models of effect of GII on the probability of emotional Isolation (SNI_emo_ < 3).

	Model D				Model C	
Predictor	Estimate	SE	Sig.	Estimate	SE	Sig.
Intercept	−0.958	0.303	***	−1.3	0.231	***
|logGII|	0.193	0.122		0.205	0.105	*
Controls	Yes			No		
SD Country	0.448	0.005		0.439	0.001	
ICC adjusted	0.058			0.055		
ICC conditional	0.052			0.055		
LRT-Test	Chisq = 3168.91 DF = 20 Pr(>Chisq) < 0.001			Chisq = 3.43 DF = 1 Pr(>Chisq) < 0.064		

Signif. codes: “***” *p* < 0.001, “*” *p* < 0.05.

**Table 6 ijerph-19-07428-t006:** Multilevel linear models of the effect of GII on SNI_community_.

	Model II				Model I	
Predictor	Estimate	SE	Sig.	Estimate	SE	Sig.
Intercept	0.037	0.066		0.163	0.058	***
|logGII|	0.041	0.027		0.045	0.027	*
Controls	Yes			No		
SD Country	0.102	0.001		0.11	<0.001	
SD Observation	0.475	<0.001		0.479	<0.001	
ICC adjusted	0.044			0.05		
ICC conditional	0.043			0.05		
LRT-Test	Chisq = 839.05 DF = 20 Pr(>Chisq) < 0.001			Chisq = 2.98 DF = 1 Pr(>Chisq) < 0.084		

Signif. codes: “***” *p* < 0.001,“*” *p* < 0.05.

## Data Availability

Not applicable.

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
