# Peer review of "Loneliness and Emancipation: A Multilevel Analysis of the Connection between Gender Inequality, Loneliness, and Social Isolation in the ISSP 2017"

_ijerph, 2022, doi:10.3390/ijerph19127428_

Round 1

Reviewer 1 Report

The table added with country-level descriptive statistics includes the means and standard deviations of descriptive measures. It should also include the total N or how many people responded from each country to aid readers in knowing the sample size representing each country.

Discussion is much improved.

Should define ISSP in the abstract before moving to using the acronym for the rest of the document.

Author Response

Dear Reviewer,

Thank you very much for taking the time to review my paper “Loneliness and Emancipation” for a second time. 

Regarding your comments:

I added the acronym in the abstract and added a column with the country level N to the table with the descriptive statistics in the appendix. 

Also, I am very pleased that you find the discussion much improved. I thank you again for your comments in the first round of reviews that made this possible.

This manuscript is a resubmission of an earlier submission. The following is a list of the peer review reports and author responses from that submission.

Round 1

Reviewer 1 Report

This is a very interesting paper. The author hypothesizes that more gender equal societies benefit from an emancipation dividend that reduces overall levels of loneliness. The theoretical foundation of the paper was well conceptualised and argued, and the analyses were well executed. 

I make the following suggestions for how the paper might be improved:

1. In theorising associations between gender equality and loneliness, you focus on the effects of increasing access to divorce - only the high-quality relationships will survive. I agree that this is likely to be a key mechanism. However, is it not also possible that increased gender equality leads to more supportive, emotionally close, egalitarian and hence satisfying relationships between men and women? So, not only are poorer-quality relationships less likely to survive, but there are fewer of those relationships to begin with? I think you allude to this in a couple of places, but I would like to see you address it more explicitly.

2. In theorising the associations between gender equality and social isolation, I believe that you have overlooked an important point.  Women's social isolation arguably decreases substantially when their workforce participation increases. In more gender equal societies, there are fewer women home alone with young children or in an 'empty nest'. I would expect that increased workforce participation more than compensates for decreased volunteering or community/church engagement. Workplaces are often an important source of social contact and even emotional support, as evidenced by this quote from a 1997 study by Hochschild:

"Often, working parents feel more at home at work because they come to expect that emotional support will be more readily available there. As at Amerco, work can be where their closest friends are, a pattern the Bright Horizons survey reflected. When asked, "Where do you have the most friends?" 47 percent answered "at work"; 16 percent, "in the neighborhood"; and 6 percent, "at my church or temple." Women were far more likely than men to have the most friends at work." (From: Hochschild, Arlie (1997). "When work becomes home and home becomes work", California Management Review, vol. 39, no. 4, pp. 79-97).

I would contend that this is even more likely to be the case in the most gender equal countries, such as the Nordic countries, where there is more support for both women and men to care for children and parents experience less work-family conflict as a result. I would expect that this leads to an improvement in the quality of relationships in both the workplace and the home (this also links to my first point).

3. One important possibility that you haven't addressed is that the effects of increasing gender equality on loneliness may be heterogeneous. For example, it seems plausible that the reduction in loneliness would be more pronounced for women than for men. However, effects might also vary across other aspects of social identity such as SES/class, race/ethnicity, religious affiliation etc. For example, is the emancipation dividend more of a "middle class luxury" that poorer, working-class women can't afford? Further, are there groups in the population who 'lose out' when gender equality increases - such as undesirable (violent, mediocre) men, who no doubt find it easier to 'trap' a woman in marriage in less gender equal societies? I'm thinking here of the incel movement. While it may be beyond the scope of your current study to investigate this, I think it is something you should discuss in your concluding section as an important avenue for future research that builds on the current study.

4. I was disappointed in the brevity of your discussion. There are several interesting avenues for further research that spring to mind (such as that mentioned in my previous point), which you could raise here. Further, you should outline your study's limitations, and discuss other plausible interpretations for your findings.

Author Response

Response to Referee Report #1 from IJERPH

Dear Reviewer,

Thank you very much for taking the time to give me guidance on improving my paper “Loneliness and Emancipation”. I value your thoughtful comments highly. I tried to incorporate them into the text and respond to them as fully as possible. You will find my direct responses to each suggestion (in italics) below. I worked through all of the comments and remarks, and I think that my paper has improved significantly as a result of following them. I have revised the paper substantially: some parts have been rewritten or supplemented, and others have been dropped, in all cases following your and the other referee’s advice.

Comment 1:

This is a very interesting paper. The author hypothesizes that more gender equal societies benefit from an emancipation dividend that reduces overall levels of loneliness. The theoretical foundation of the paper was well conceptualised and argued, and the analyses were well executed. 

Thank you very much for your appreciation.

Comment 2:

I make the following suggestions for how the paper might be improved:

  1. In theorising associations between gender equality and loneliness, you focus on the effects of increasing access to divorce - only the high-quality relationships will survive. I agree that this is likely to be a key mechanism. However, is it not also possible that increased gender equality leads to more supportive, emotionally close, egalitarian and hence satisfying relationships between men and women? So, not only are poorer-quality relationships less likely to survive, but there are fewer of those relationships to begin with? I think you allude to this in a couple of places, but I would like to see you address it more explicitly.

This is an excellent point. I tend to agree with it and included a discussion of this argument in the text beginning from line 142 of the revised manuscript. However, when I researched this point for the revision, I noticed that the available literature, while it points in the direction of your suggestion, is very complex and often contradictory. There are also lots of gaps in it. For instance, I couldn’t find papers based on longitudinal data that controlled for initial (in-)equality in the relationship (which could be interpreted as a selection effect due to equality norms) to distinguish selection from relationship equality effects. For this reason, I kept the discussion in the text rather short.   

Comment 3:

 In theorizing the associations between gender equality and social isolation, I believe that you have overlooked an important point.  Women's social isolation arguably decreases substantially when their workforce participation increases. In more gender-equal societies, there are fewer women home alone with young children or in an 'empty nest'. I would expect that increased workforce participation more than compensates for decreased volunteering or community/church engagement. Workplaces are often an important source of social contact and even emotional support, as evidenced by this quote from a 1997 study by Hochschild:

"Often, working parents feel more at home at work because they come to expect that emotional support will be more readily available there. As at Amerco, work can be where their closest friends are, a pattern the Bright Horizons survey reflected. When asked, "Where do you have the most friends?" 47 percent answered "at work"; 16 percent, "in the neighborhood"; and 6 percent, "at my church or temple." Women were far more likely than men to have the most friends at work." (From: Hochschild, Arlie (1997). "When work becomes home and home becomes work", California Management Review, vol. 39, no. 4, pp. 79-97).

I understand this point and include its discussion in the paper starting from line 254 in the revised manuscript. However, I tend to disagree with a diachronic interpretation of Hochschild's findings. This is a rather complex topic. From my point of view, at its center is a conceptual issue: Close friends in more traditional and also in most historic societies are chosen dominantly among family and kin. This is one reason the friendships of women are very often overlooked in historic analyses of friendships, because they are simply assumed to be family or kinship relationships. Where kinship structures start to become scarcer, this situation shifts: Friends are now chosen more often among non-kin such as workplace acquaintances, acquaintances from school, etc. This can lead to the impression that women suddenly are having more friends when their labor participation increases. But this is for the most part an effect of applying a friendship concept that defines friends as non-kin, which is a very modern, western way of looking at it. In short: I have strong doubts that the increase in female labor participation has a substantial effect on the number of friendships and acquaintances women form.

Comment 4:

I would contend that this is even more likely to be the case in the most gender equal countries, such as the Nordic countries, where there is more support for both women and men to care for children and parents experience less work-family conflict as a result. I would expect that this leads to an improvement in the quality of relationships in both the workplace and the home (this also links to my first point).

This argument I consider more likely to hold up to strong scrutiny. Beyond what you suggest, I would expect that relationship stability and mobility norms also affect friendship formation in more gender-equal societies. There are some empirical correlates that fit with this idea. For instance, the gender homogeneity of close friendships is especially high in Norway but rather low in Chile in the data of the ISSP 2001. I never published this result, but what it suggests to me is that more modern societies offer less constraints to friendship formation because friends are chosen more often outside of family and kinship. An increase in the freedom to choose and dissolve friendships should translate into an increase in homogeneity but also in higher friendship quality, for similar reasons to that of the increase in the quality of marriages. However, debating this would be stuff for a whole additional paper, so I did not include a discussion of it.

Comment 5:

One important possibility that you haven't addressed is that the effects of increasing gender equality on loneliness may be heterogeneous. For example, it seems plausible that the reduction in loneliness would be more pronounced for women than for men. However, effects might also vary across other aspects of social identity such as SES/class, race/ethnicity, religious affiliation etc. For example, is the emancipation dividend more of a "middle class luxury" that poorer, working-class women can't afford? Further, are there groups in the population who 'lose out' when gender equality increases - such as undesirable (violent, mediocre) men, who no doubt find it easier to 'trap' a woman in marriage in less gender-equal societies? I'm thinking here of the incel movement. While it may be beyond the scope of your current study to investigate this, I think it is something you should discuss in your concluding section as an important avenue for future research that builds on the current study.

This is a very good point. I now discuss avenues for further research in my paper with more detail and included your suggestions prominently among the things that merit further investigation. I included your remark prominently among the things that should be investigated further from line 694 onwards

Comment 6:

I was disappointed in the brevity of your discussion. There are several interesting avenues for further research that spring to mind (such as that mentioned in my previous point), which you could raise here. Further, you should outline your study's limitations, and discuss other plausible interpretations for your findings.

I added substantially to the discussion by reflecting more explicitly on the limitations of my research and about avenues for further research

Reviewer 2 Report

The authors base much of their discussion on the implications of how gender equality impacts numbers of, closeness of, and stress in close relationships across countries. I believe the paper would benefit from a more explicit literature review and demonstration of the causal relationship between gender equality and these variables, rather than simply discussion correlational data in terms of a presumed causality that is not explicitly demonstrated. I also found that the number of variables controlled for in the final analysis to be quite large, and believe that if the paper's final conclusion rests on a model including all of these variables then those items should be more thoroughly discussed sooner and a theoretical model should be built such that the reader has a clear picture of the overall model being examined. Perhaps a chart or graphic representation earlier in the paper would be helpful. In particular, depressive symptoms seem like they would be quite redundant with loneliness, and it would be nice to know how correlated those variables generally are. Also, the total sample size for all 30 countries was reported but the authors should note how many people were surveyed in each country as well as supplemental any differences between national surveys. Finally, the authors should spend a bit of time discussing the implications of this work for both policy at the national level and overall quality of life for citizens.

The authors overall investigation is interesting, and I believe with clarification of the above would be more compelling.

Author Response

Response to Referee Report #2 from IJERPH

Dear Reviewer,

Thank you very much for taking the time to give me guidance on improving my paper “Loneliness and Emancipation”. I value your thoughtful comments highly. I tried to incorporate them into the text and respond to them as fully as possible. You will find my direct responses to each suggestion (in italics) below. I worked through all of the comments and remarks, and I think that my paper has improved significantly as a result of following them. I have revised the paper substantially: some parts have been rewritten or supplemented, and others have been dropped, in all cases following your and the other referee’s advice.

Comment 1:

I believe the paper would benefit from a more explicit literature review and demonstration of the causal relationship between gender equality and these variables, rather than simply discussion correlational data in terms of a presumed causality that is not explicitly demonstrated.

I am a bit confused by the generality of this critique. As I see it, the causalities can never be “explicitly demonstrated” in observational studies. This bar is too high for any observational research and most sociological research necessarily is. What I would concede, however, is that the causalities that I assume to be at work could be established better with longitudinal cross-national, panel data or by data that contains suitable instrumental variables (“natural experiments”). For now, the ISSP 2017 is among the best cross-national data available, and unfortunately, it's only cross-sectional. So, I concur that my paper only goes a first step towards establishing the causalities I see at work by testing if the correlational patterns observed are consistent with the predictions of my theoretical interpretations. I now address this issue explicitly in a new paragraph on limitations from line 618 onwards.  

Comment 2:

I also found that the number of variables controlled for in the final analysis to be quite large, and believe that if the paper's final conclusion rests on a model including all of these variables, then those items should be more thoroughly discussed sooner and a theoretical model should be built such that the reader has a clear picture of the overall model being examined.

I do not understand this point very well: The comment seems to imply that I follow a ‘kitchen-sink approach’ by just putting all sorts of variables in the model. But this is not the case. The control variables were selected on theoretical grounds. They have all been linked by studies to loneliness or – such as gender and age are routinely included in statistical analyses to rule out obvious sample composition effects due to different sampling strategies such as stratification by age. I tried to clarify this issue by expanding on it a bit from line 473 onwards. Furthermore, I compare a model with and without the control variables. This is explicit in the text: Where there are differences in the results between the model with and without the controls, I discuss this in the paper (e.g. lines 560- 580).

Comment 3:

Perhaps a chart or graphic representation earlier in the paper would be helpful.

I do like charts and graphics and appreciate that charts and graphics could improve the readability of the paper. However, I can’t gather from this comment what the chart or graphic should try to represent. The causal links are expected between gender equality and the dependent variables? The connections between the control variables and loneliness and social isolation? With some further clarification, I could try to include a chart or a graphic to improve the manuscript.

Comment 4:

In particular, depressive symptoms seem like they would be quite redundant with loneliness, and it would be nice to know how correlated those variables generally are.

I do not agree with this comment. It has been established by a plethora of research that depressiveness and loneliness are distinct psychological phenomena. The view I share is that there is a significant level of “comorbidity” between loneliness and depressiveness, which needs to be and currently is being researched. The correlation of both variables, thus, does not make them redundant from where I am standing. The data supports this claim: In the ISSP the variables used only have a correlation of r=0.405, which is a moderate level of correlation. I also checked for collinearity issues estimating a linear fixed effect regression with loneliness as a dependent variable and all control variables excluding the country-level effects. Here, the Variance Inflation Factor of the depressiveness variable is low and unproblematic (1.140). In short, I do not think that the issue raised is a problem for my analysis on a conceptual or a methodological level.

Comment 5:

Also, the total sample size for all 30 countries was reported but the authors should note how many people were surveyed in each country as well as supplemental any differences between national surveys.

Table 1 now includes the country N. Also, I included a table with further country-level descriptives in Appendix A and added a reference to an in-depth analysis of the country-level results of the ISSP 2017 of Hadler et al 2020 in line 436 that also touches on the differences in survey methodology between countries, which is also discussed amply by the technical reports on the ISSP that are publicly available.  

Comment 6:

Finally, the authors should spend a bit of time discussing the implications of this work for both policies at the national level and the overall quality of life for citizens.

I now discuss the implications of the work for social policy from line 664 onwards

Comment 7:

The author's overall investigation is interesting, and I believe with clarification of the above would be more compelling.

Thank you very much for the appreciation

Reviewer 3 Report

The authors have presented an interesting study about this topic. There is some valuable and engaging material, but it should be analysed according to the findings in the Discussion section. Therefore, I believe that this manuscript would benefit from a more developed discussion and conclusion section. I suggest that the authors revise this section (and consider separating them), to allow a better alignment with the implications of the results.

I also recommend another read of the manuscript to edit some minor spelling mistakes (e.g. Baumann instead of Bauman – line 74) and consider rephrasing some sections to improve readability (e.g. Line 48 - The sentence that starts here is quite long and its style feels detached from the rest of the manuscript).

Author Response

Response to Referee Report #3 from IJERPH

Dear Reviewer,

Thank you very much for taking the time to give me guidance on improving my paper “Loneliness and Emancipation”. I value your thoughtful comments highly. I tried to incorporate them into the text and respond to them as fully as possible. You will find my direct responses to each suggestion (in italics) below. I worked through all of the comments and remarks, and I think that my paper has improved significantly as a result of following them. I have revised the paper substantially: some parts have been rewritten or supplemented, and others have been dropped, in all cases following your and the other referee’s advice.

Comment 1:

The authors have presented an interesting study about this topic.

Thank you very much for your appreciation.

Comment 2:

There is some valuable and engaging material, but it should be analysed according to the findings in the Discussion section. Therefore, I believe that this manuscript would benefit from a more developed discussion and conclusion section. I suggest that the authors revise this section (and consider separating them), to allow a better alignment with the implications of the results.

I extended the discussion and conclusion section by going into more detail regarding the limitations of my study and its potential policy implications

Comment 3:

I also recommend another read of the manuscript to edit some minor spelling mistakes (e.g. Baumann instead of Bauman – line 74) and consider rephrasing some sections to improve readability (e.g. Line 48 - The sentence that starts here is quite long and its style feels detached from the rest of the manuscript).

The manuscript has been subjected to another round of professional proofreading and I shortened lengthy sentences throughout the manuscript to make it more readable.